# The Prevalence of Orthorexia Nervosa among Greek Professional Dancers

**DOI:** 10.3390/nu15020379

**Published:** 2023-01-12

**Authors:** Dafni Athanasaki, John Lakoumentas, Gavriela Feketea, Emilia Vassilopoulou

**Affiliations:** 1Department of Nutritional Sciences and Dietetics, International Hellenic University, 57400 Thessaloniki, Greece; 2Department of Hematology, “luliu Hatieganu” University of Medicine and Pharmacy, 400337 Cluj-Napoca, Romania; 3Department of Pediatrics, “Karamandaneio” Children’s Hospital of Patra, 26331 Patra, Greece

**Keywords:** orthorexia nervosa, body image, body mass index, parental bonding relationship, professional dancers

## Abstract

The aim of this study was to determine the prevalence of orthorexia nervosa (ON) among professional dancers in Greece, as well as its relationship with nutrition, body mass index (BMI), body image flexibility, and parental bonding. The participants were 96 professional dancers, with a mean age of 23.41 ± 5.13 years, who completed a battery of questionnaires recording sociodemographic, clinical, and anthropometric characteristics; adherence to the Mediterranean diet (MedDiet); indications of ON, as determined by the ORTHO-15 questionnaire; body image flexibility, using the body image-acceptance and action questionnaire (BI-AAQ-5); and their recollection of their parents’ attitudes towards them during the first 16 years of life, with the parental bonding instrument (PBI). The study population was classified into two groups, based on BMI: normal weight, and underweight. ON was shown to be significantly correlated with BMI (*p* = 0.006)-present in normal weight subjects- and body image inflexibility (*p* < 0.001). Parental body image inflexibility was significantly correlated with a low bonding relationship as perceived in childhood. In conclusion, disordered eating attitudes and body shape concerns are prevalent among professional dancers and appear to be associated with their parental relationship during childhood. Identification of potential ON and development of preventive mechanisms could help to eliminate such concerns and improve the nutrition of professional dancers.

## 1. Introduction

Orthorexia nervosa (ON) is a term that refers to an unhealthy insistence on healthy eating, and was first used by Bratman [1]. According to his definition, a healthy diet evolves into ON when a person crosses a “borderline” beyond which his/her concern about food begins to exert a negative effect on essential areas of life, including psychological, physical, and social aspects [2]. It is often difficult to define the point of transition, but ON can be described as a state in which the pursuit of a healthy diet dominates the individual’s life and ceases to serve the purpose of improving health [3,4]. Although, to date, no formal diagnosis of ON has been included in the Diagnostic and Statistical Manual for Mental Disorders (DSM-5) definitions of eating disorders (ED) [2,5], a number of diagnostic criteria have been proposed [6,7,8]. There is some variation between criteria, but the following characteristics are commonly described: an obsession with eating healthy or pure foods, repressive behavior or mental preoccupation in relation to foods considered unhealthy, and emotional distress and fear linked to food perceived as unhealthy and its potential harmful effects on the body and health. Consequently, several complications may appear, such as nutritional deficiencies due to elimination of entire food groups, severe weight loss, or other medical complications. The recurrent need to maintain control of healthy eating may cause self-exclusion from social contexts, resulting in educational or occupational impairment and social isolation [9]. While weight loss was not described in the original reports as a goal of those with ON tendencies, there is increasing recognition that some people adopt ON behaviors because, in contrast to people with anorexia nervosa (AN), whose main aim is to lose weight, they are seeking to achieve an ideal image of health/wellness and “correct” weight, or simply because of confusion between “healthy” and “low calorie” foods [2].

Cheshire and colleagues (2020) identified a number of features of the parental influence on ON tendencies, including “nurturing issues”, related to parents being absent or emotionally unavailable [2]. A challenging relationship with parents was proposed as a factor leading to controlled eating behavior in offspring [2]. In addition, exposure in childhood and adolescence to extreme parental attitudes and behaviors, particularly those related to exercise and nutrition, appear to have a general impact on the attitude to food and health of young people [2].

Lobera and colleagues (2011) [10] emphasized the critical role of parents in helping adolescents and early adults to master the tasks of this period in their development. Several studies have suggested that a particular style of parenting behavior, namely authoritative parenting, is advantageous for supporting adolescents. Parents who adopt this style are warm and involved, and employ non-punitive disciplinary methods to support healthy development and self-acceptance in their children. In contrast, punitive parents may apply either a distant and hierarchical authoritarian style or neglectful/abusive parenting. Ideally, parents should be involved, but should be firm and consistent in establishing and enforcing guidelines and limits, as well as having developmentally appropriate expectations of their children [10]. Based on the above, parental involvement could facilitate the prevention and treatment of eating disorders (ED) in adolescents and young adults, including ON. Lobera and colleagues (2011) reported that a stereotypical parenting style during the first 16 years, characterized by low care and high control, was common among patients with ED [10]. A relationship between a low level of parental care and psychopathological symptoms, weight phobia, and body image has been documented [10]. It was estimated that 8.6–12.9% of patients considered their parents’ style to be neglectful. On the relevant questionnaire, patients who perceived their mothers as neglectful during their first 16 years recorded higher scores on the subscales of drive for thinness and body dissatisfaction than those who perceived affectionless control and affectionate constraint styles in their mothers [10].

Some studies mentioned that dancers are three-times more likely than their non-dancer counterparts to develop an ED. Kulshreshtha and colleagues (2020) found that approximately 83% of dancers reported body shape concerns, ED behaviors, or the use of a non-healthy weight regulation strategy [11]. Many dancers have a distorted body image, linked with the induction of ED [12] and the development of body dissatisfaction [13]. It has been documented that, given the esthetic nature of dance, a thin body is often idealized and normalized in the dance culture. This would potentially result in body shape concerns, with a high rate of ED risk factors (e.g., internalization of a thin ideal of beauty), and the development of psychopathology (e.g., compulsive exercise) [12,13]. For the purpose of maintaining or reducing their weight when aiming for the thin “ideal”, dancers may start excessive training or develop disordered eating habits [11].

The aim of this study was to examine the factors that may lead to body image inflexibility and ON in dancers, with an emphasis on the role of parental bonding and adherence to a healthy Mediterranean diet (MedDiet).

## 2. Materials and Methods

### 2.1. Participants and Procedures

A retrospective, observational study was conducted on professional dancers in Greece. A total of 100 individuals were recruited through purpose sampling from professional dance schools and dance workplaces. Of these, three were excluded due to a non-professional relationship with dance, and one declined to participate. The remaining 96 participants were professional dancers, who were either studying at, or had graduated from, a higher professional dance school and who worked as dancers or dance teachers. The study was conducted with the permission of the Committee for Research Ethics of Aristotle University of Thessaloniki (1.272/20.10.2020) and in accordance with the code of Ethics of the World Medical Association (Declaration of Helsinki). All the participants were fully informed of the scope and procedures of the study and provided written consent.

### 2.2. Sociodemographic, Medical, and Anthropometric Characteristics

A personal interview was conducted with each participant by a member of the research team, to collect self-reported information on the following: (a) anthropometric characteristics, including age, body weight, and height. Body mass index (BMI, kg/m^2^) was calculated from the participants’ self-reported height and weight, and categorized using the World Health Organization (WHO) BMI classification (>18.5: underweight; 18.5–24.9: normal weight; 25.0–29.9: pre-obesity; 30.0–34.9: obesity class I; 35.0–39.9: obesity class II; >40: obesity class III) [14]; (b) socio-demographic and occupational information, including level of education, level of dance education and professional dance level, total years and daily duration of performing dance, and other frequent physical activity; (c) medical history, including chronic disease, and medication and use of dietary supplements. Specific tools were used to measure diet, orthorexia, body image, and parenting, as listed below.

#### 2.2.1. Dietary Habits in Relation to Adherence to the Mediterranean Diet

Dietary habits were evaluated through assessment of adherence to the MedDiet. For this purpose the validated MedDiet Score questionnaire was used [15]. Eleven main MedDiet components are listed in this questionnaire, for which the participants report the frequency of consumption. A MedDiet composite score was calculated as follows: for the components commonly consumed in the MedDiet (non-refined cereals, potatoes, fruits and fruit juice, vegetables, and salad, legumes, and fish), a score of 0 (lowest adherence) to 5 (highest adherence) was assigned, for reported consumption of 0, 1–4, 5–8, 9–12, 13–18, and >18 servings/month, respectively. For the components which are less frequently consumed in the MedDiet (red meat and meat products, poultry, and whole-fat dairy products), a score of 0–5 was assigned for reported consumption, using a reverse scale. A score of 0–5 was assigned for using olive oil (in both meal preparation and cooking) ‘never’, ‘rarely’, ‘<once/week’, ‘1–3 times/week’, ‘3–5 times/week’, and ‘daily’, respectively. For alcohol (all alcoholic beverages), a score of 5 was assigned for ‘no consumption’ or ‘consumption of <300 mL/day’, a score of 0 for ‘consumption of >700 mL/day’, and scores of 4 to 1 for the consumption of ‘300–400 mL’, ‘400–500 mL’, ‘500–600 mL’, and ‘600–700 mL/day’, respectively. The resulting total score ranged from 0–55, with calculated quartiles indicating low (0–20), moderate (21–35), and high (36–55) adherence to the MedDiet [15,16].

#### 2.2.2. Orthorexia

The ORTHO-15, a self-administered questionnaire made up of 15 items on a four-point Likert scale (1 = always, 2 = often, 3 = sometimes, 4 = never), was used to assess the incidence of ON in the study sample [9]. The questionnaire investigated three underlying factors of eating behavior: cognitive–rational (explored by questions 1, 5, 6, 11, 12, 14), clinical (questions 3, 7, 8, 9, 15), and emotional (questions 2, 4, 10, 13). A lower score indicates a higher level of ON symptomatology. The authors suggest a cut-off point of 40 for the diagnosis of ON [3].

#### 2.2.3. BIAAQ

Body image flexibility was assessed through the Body Image-Acceptance and Action Questionnaire 5 (BI-AAQ-5). Body image flexibility is correlated with increased psychological flexibility, decreased body image dissatisfaction, and less disordered eating [17]. The BI-AAQ was adapted from existing measures of psychological flexibility, to specifically assess the response to body-related thoughts and feelings [17]. The BI-AAQ-5 [18] is an abbreviated version of the original measure and consists of five items, rated on a scale ranging from 1 (never true) to 7 (always true) [19]. According to Basarkod and colleagues [18], the short form performed comparably to the long form, in terms of its factor structure and correlation with theoretically relevant constructs, including body image dissatisfaction, stigma, internalization of societal norms of appearance, self-compassion, and poor mental health. The items are negatively worded, thus requiring reverse scoring of each item before summing all the items to determine the total score [19]. Those who score 19 or above are potentially at risk of distorted body image [18]. The participants also completed the questionnaire in relation to the perceived body image flexibility of their own parents.

#### 2.2.4. Parental Bonding Instrument

The Parental Bonding Instrument (PBI) was completed for both parents, to retrospectively evaluate how the participants remembered their parents during their first 16 years of life. Briefly, it is comprised of two scales to measure care and overprotection and control, as fundamental parental styles as perceived by the child, and was designed to be completed for mothers and fathers separately. It consists of 25 items, including 12 care items and 13 overprotection items. The classification into high or low categories is in accordance with the following cut-off scores: for mothers, a care score of 27.0 and a protection score of 13.5; for fathers, a care score of 24.0 and a protection score of 12.5. These cut-off points discriminate between four different styles: affectionless control (high control–low care), optimal (low control–high care), affectionate constraint (high care–high control) and neglectful parenting (low care–low control) [10].

### 2.3. Statistical Analysis

SPSS Software version 21.0 was used to perform statistical analysis. All the score variables, along with BMI, were correlated with each other, in both continuous and discretized form. Specifically, the discretized forms were assessed for pairwise associations using Pearson’s chi-squared test of independence, and the continuous forms were assessed for pairwise correlations with Pearson’s R-correlation test. To discriminate the factors that affected ON, the ORTHO-15 outcome was set as the dependent variable (target), and all other variables as independent (predictors). All of the demographic characteristics were encoded as categorical variables, either originally discrete or discretized (for instance the BMI scale was used to discriminate participants into under- or normal-weight) and were described as ‘absolute count (relative percent)’ per categorical level, apart from age, which was normally distributed and described as ‘mean ± standard deviation’. Pearson’s chi-squared independence test was used to assess dependencies of the categorical predictors with the target, and age was assessed for dependency with the target using Student’s (unpaired) t-test. *p*-values equal to or less than 0.05 (*p* ≤ 0.05) were considered statistically significant. Subsequently, the sample was divided based on BMI into two groups, underweight and normal weight, and the analysis was repeated according to BMI groups. Beyond hypothesis testing in all cohorts, a binary logistic regression procedure (in a multivariate manner) was used to either validate or disprove the hypothesis testing findings.

## 3. Results

Among the 96 professional dancers who participated in the study, 92 were women (95.8%), 3 were men (3.1%), and 1 other gender (1.0%). The mean age of the participants was 23.41 ± 5.13 years, and 84 were of normal weight (87.5%) and 12 were underweight (12.5%).

Of the 96 participants, 13 (13.7%) had graduated from a professional dance school, 81 (85.3%) were attending a professional dance school, and one (1.1%) worked as a dancer/dance teacher. The majority had been involved with dance for 10 or more years (*n* = 73, 76.8%); 46 (48.4%) performed dance 5–6 times/week and 41 (43.2%) performed dance daily. Most of them danced daily for 3–4 h/day (36, 37.9%) or more than 4 h/day (41, 43.3%).

Αpart from dance, some participants performed an additional physical activity, specifically 12 (12.5%) aerial acrobatics, 9 (9.4%) pilates, 6 (6.3%) exercise with weights, 5 (5.2%) running, 3 (3.1%) yoga, 3 (3.1%) rhythmic gymnastics, 2 (2.1%) cycling, 2 (2.1%) climbing, and one (1.0%) each, basketball, tennis, skating, kickboxing, volleyball, and swimming.

Concerning history of chronic disease, four (4.2%) participants had a history of Hashimoto’s thyroiditis, three (3.1%) hypothyroidism, and one (1.0%) of hypercholesterolemia. In addition, several participants were taking medication: nine (9.4%) regularly took T4 (levothyroxine sodium), two (2.1%) antidepressants, one (1.0%) antihypertensives, one (1.0%) drugs for hypercholesterolemia, one (1.0%) an anti-TNFα drug, and one (1.0%) contraceptives.

Some participants reported following a specific diet; 11 (11.5%) a lacto-ovo-vegetarian diet, and one (1.0%) a lacto-ovo-fish vegetarian diet.

Concerning MedDiet, overall, the participants presented a medium degree of adherence to the MedDiet (score: 32.66 ± 5.60), with 4 (4.2%) having high, 73 (76%) medium, and 19 (19.8%) a very low score.

Regarding dietary supplements, 16 (16.7%) used magnesium, 13 (13.5%) a multivitamin, 10 (10.4%) iron, 9 (9.4%) vitamin C, 9 (9.4%) vitamin D, 5 (5.2%) vitamin B, and 2 (2.1%) a protein supplement. A variety of other supplements were individually used by one participant (1% each), including zinc, melatotin, potassium selenate, branched chain amino acids (BCAA), glucosamine, melatonin, electrolyte, omega-3, potassium, selenium, chondroitin, and cranberry supplement.

### 3.1. Risk of Orthorexia Nervosa

According to the scores on the ORTHO-15 questionnaire, 71 (74%) presented symptoms of ON, with a mean score of 35.75 ± 5.69, among whom 84 (87.6%) were of normal weight and 12 (12.5%) were underweight, with a mean BMI of 20.19 ± 1.69. As revealed by the scores on the BI-AAQ-5, body image inflexibility was present in 64 (66.7%) participants, with a mean score of 21.15 ± 7.99. As shown in Figure 1, 80.3% of the participants with ON, and only 28% of those without ON showed body image inflexibility. Of the participants with ON, 93% were of normal weight and 7% were underweight. The majority of the participants with ON (*n* = 58, 81.7%) were students at a professional dance school.

Table 1 presents the factors associated with ON, as assessed by the ORTHO-15 questionnaire. ON was strongly associated with body image inflexibility, as self-reported with the BI-AAQ-5 (*p* < 0.001). Specifically, the participants with a risk of body image inflexibility, scored higher for ON (*p* < 0.001), while BMI was a significant confounding variable (*p* = 0.006). No other parameter emerged as a significant predictor of ON.

To eliminate the effect of BMI, we separated the participants according to their BMI into two categories: underweight (12.5%; *n* = 12) and normal weight (87.5%; *n* = 84), and repeated the analysis for each group separately. In the underweight group, ON was not associated with body image inflexibility. In the normal weight group, as shown in Figure 2, a higher degree of body image inflexibility was associated with an increase in the symptoms of ON (*p* < 0.001). In this group, 66/84 were classified as ON, with 54 (81.8%) having body image inflexibility, while 18/84 were non-ON, with 5 (27.8%) having body image inflexibility.

To examine the relationship between ON, body image flexibility and parental bonding relationship, parental PBI was set as the dependent variable; no statistically significant association was revealed. Similarly, the MedDiet score as a dependent variable did not reveal significant associations.

### 3.2. Association between Orthorexia Nervosa, Body Image Inflexibility, and Parental Bonding Profiles

With regard to maternal bonding during the first 16 years of life, as evaluated with the PBI, 12 (12.5%) participants had received affectionate constraint, 54 (56.3%) optimal, 12 (12.5%) affectionless control, and 17 (17.7%) neglectful parenting, while 1 (1.0%) person grew up without a mother. With regard to paternal bonding, 4 (4.2%) participants received affectionate constraint, 45 (46.9%) optimal, 18 (18.8%) affectionless control, and 21 (21.9%) neglectful parenting, while 8 (8.3%) participants grew up without their father.

Among the 71 participants without ON, the corresponding figures for maternal bonding were as follows: 9 (12.7%) had received affectionate constraint, 39 (54.9%) optimal, 9 (12.7%) affectionless control, 13 (18.3%) neglectful parenting, and 1 (1.0%) grew up without a mother. Paternal styles were 3 (4.2%) with affectionate constraint, 29 (40.8%) optimal, 16 (22.5%) affectionless control, 16 (22.5%) neglectful parenting, and 3 (4.2%) grew up without their father.

Of the 30 participants who reported a low maternal bonding score (neglectful parenting; affectionless control), 23 (76.7%) scored high for orthorexia. Of the 47 participants with low paternal bonding, 39 (83%) had ON.

Poor maternal bonding was reported in a subgroup of 21 participants characterized by both body image inflexibility and a substantial risk of ON (81%). In another subgroup of 35 participants, poor paternal bonding was co-present with body image inflexibility and 88.6% (31/35) also had a high orthorexia score.

### 3.3. Association between Parental Body Image Inflexibility and Parental Bonding Profiles

Parental score levels on BI-AAQ were strongly associated with the maternal and paternal scores on PBI, as perceived by the participants. Specifically, there was a lower risk of parental body image inflexibility when the parental bonding was high, especially when optimal nurturing was provided by both parents (*p* < 0.001 for both parents).

## 4. Discussion

This study assessed the risk of ON among professional dances in relation to body image flexibility, BMI, parental bonding, and MedDiet score.

Healthy eating habits were evaluated according to the adherence of the dancers to the MedDiet, and it was found that 80.4% had moderate to high adherence, in contrast with previous studies [20] that reported poor adherence of dancers to the MedDiet. The MedDiet has long been recognized to be associated with a low incidence of cardiovascular disease and cancer [15]. Our findings are consistent with other studies, which described the eating behaviors of dancers as healthy, with regular meals and a wide variety of foods from all food groups, including a high intake of fruit and vegetables, yoghurt, and kefir, along with a low consumption of fast food products [21].

In our cohort of dancers, supplement use was reported by 41.4%, in line with previous studies that reported similar supplement use (48%) among dancers [22]. It is of note that, in our study, magnesium supplementation (16.7%) was the most frequent; a previous study had reported magnesium deficiency in ballet dancers [23]. Brown and colleagues (2014) reported that the most frequently consumed supplements in their sample of dancers were vitamin C (60%), multivitamins (67%), and caffeine (72%) [22]. In our cohort a multivitamin supplement was consumed by 13.5%, while 9.4% used vitamin C alone. Beck and colleagues (2014) reported that female adolescent ballet dancers are at risk of iron deficiency [24]. An iron supplement was used by 10.4% of our participants.

The majority of our study participants (87.5%) were of normal weight, according to the WHO BMI classification, in line with other studies [24,25], although Bacchi and colleagues (2012) reported that BMI was lower in both professional and non-professional dancers than in control subjects [26].

Despite their generally healthy BMI scores, 74% of the participants in our study scored high in ON symptoms. Ballet dancers, athletes, and health workers have been reported to be high risk populations for ON [9,27,28,29,30], with percentages of up to 90% [9]. It is considered that dancers may be more attuned to body image and changes in weight than members of the general population [31].

We identified a strong association between ON and body image inflexibility in the dancers. Previous findings from a Hungarian study [32] were that symptoms of ON were positively correlated with eating and body image disturbances (r = 0.46, *p* < 0.01) [32,33]. This association was also significant for the normal weight participants in our study, in contrast to findings from Dell’Osso and colleagues (2018), who reported a higher rate of ON in those with low BMI than in those with normal or high BMI (42.8% vs. 34.2%) [34]. Previous studies documented that disordered eating behaviors, attitudes, and body dissatisfaction were highly correlated among professional dancers, indicating that any increase in body shape concerns may directly affect the risk of adopting disordered eating attitudes and behaviors [11]. Camci and colleagues (2009) [30] suggested that body image dissatisfaction and forms of EDs, such as AN and bulimia, are more common in ballet dancers, as they face strong occupational pressure to be thin, and their practice of intensive exercise in combination with diet can lead to ED [30]. This strong association was confirmed by García Dantas and colleagues (2018) [35], who observed that body dissatisfaction is an important factor in the development of eating pathologies, especially among dancers.

In contrast to earlier reports, such as those of Brytek-Matera and colleagues (2014) [33], McInerney-Ernst (2011) [36], and Shah (2012) [37], our findings suggest a significant correlation between ORTHO-15 test scores and BMI, but parental influences did not emerge as a significant factor in the development of ON [2]. Lobera and colleagues (2011) [10] reported that low levels of parental care and overprotection appeared to be related to low self-esteem and a higher risk for ED [10]. In our study, parental body image flexibility, as perceived by the participants in childhood, appeared to influence both the maternal and the paternal bonding with the child. Parental body image flexibility was shown to be associated with an optimal level of care from the parents, indicating the importance of parental provision of high levels of care and affection for their children.

In this study we investigated factors with a possible influence the incidence of ON in professional dancers. One strength of our study lies in our attempt to link ON and body image flexibility with a wide range of factors, including, but not limited to, eating habits, educational level, and occupation with dance, as well as extending to the family environment in which the participants were raised. Our sample consisted exclusively of dancers with a professional involvement in dance, mostly students and graduates of higher professional dance schools approved by the Greek Ministry of Culture and Sports. To our knowledge, this is the first study to examine ON and its relationship with parental bonding among Greek professional dancers.

ON and body image acceptance disorders are complex. It was not our intention to oversimplify the issue, but to explore and present the factors that could influence the development of ON, in order to provide a deeper understanding of ON in dancers and to stimulate further study and possibly lead to intervention measures to enhance the health and well-being of this population.

The study has some limitations, the first of which is the fact that it was based on self-report instruments; the subjects may not have been able to rate themselves objectively and accurately, thus negatively affecting the validity of the results. In addition, there was no clinical assessment of ED, only an evaluation based on the subjective estimations of the study participants.

## 5. Conclusions

This study confirmed the high prevalence of ON in professional dancers, as previously reported [30], and its association with both BMI and body image flexibility. This association pertained even in subjects of normal weight, which confirms the need for subtlety in studies of subjects in the dance industry. Use of the BIAAQ indicated an inability of parents with body image inflexibility to adequately bond with their child, with implications for possible future development of ON.

The association of both BMI and body image flexibility with the development of ON helps us to understand their role in this condition that may be detrimental to a dancer’s health. Recognition of the role played by impaired parental body image and bonding, and awareness of the signs of eating disorders and body image inflexibility in professional dancers, are pathways that could lead to intervention measures to eliminate ON and improve health and well-being. Future studies should aim to explore the effect of parental bonding and early life emotional exposures in relation to ON and body image inflexibility.

## Figures and Tables

**Figure 1 nutrients-15-00379-f001:**
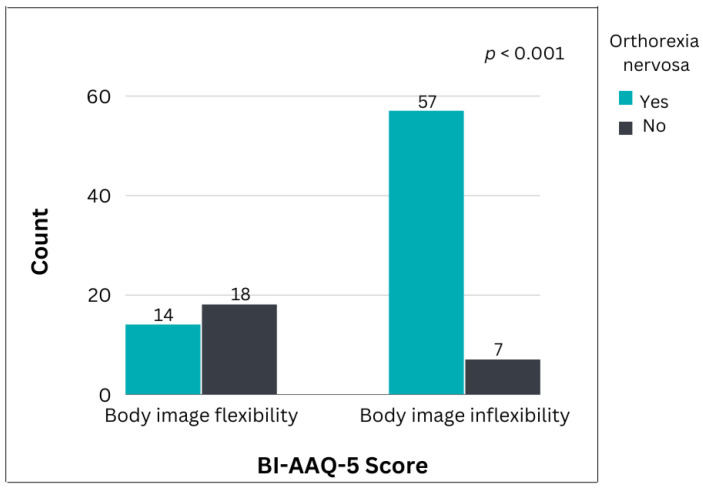
Correlation between orthorexia nervosa (ON) evaluated with the ORTHO-15 questionnaire, and body image inflexibility according to the BI-AAQ-5 score in professional dancers (*n* = 96; *p* < 0.001).

**Figure 2 nutrients-15-00379-f002:**
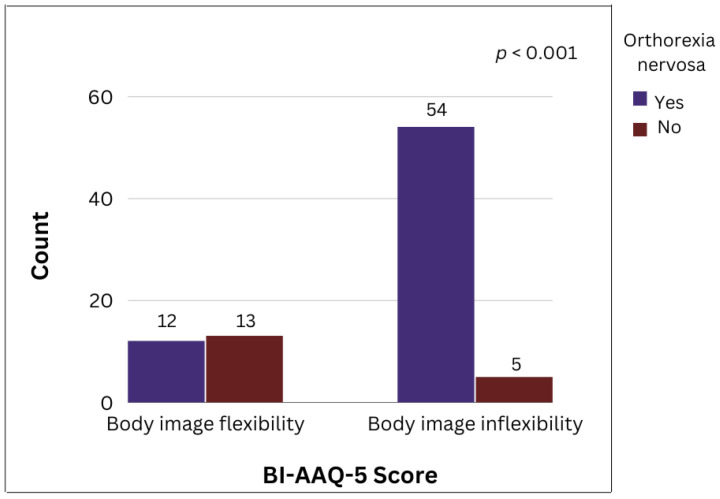
Correlation between orthorexia nervosa (ON) evaluated with the ORTHO-15 questionnaire and body image inflexibility according to the BI-AAQ-5 score in professional dancers of normal weight (*n* = 84) (*p* < 0.001).

**Table 1 nutrients-15-00379-t001:** Characteristics of professional dancers (*n* = 96), according to classification into orthorexia nervosa (ON), and non-orthorexia nervosa (non ON), based on scores in the ORTHO-15 questionnaire.

Variable	ON (*n* = 71)	Non ON (*n* = 25)	*p* Value
Gender	68 (95.8%) F ^3^/2 (2.85) M ^4^/1 (1.4%) O ^5^	24 (96.0%) F ^3^/1 M ^4^	0.805
Age (years)	23.49 ± 5.45	23.16 ± 4.2	0.782
BMI (kg/m^2^)			0.006 *
Underweight	5 (7.0%)	7 (28.0%)	
Normal weight	66 (93.0%)	18 (72.0%)	
Chronic disease	YES 7 (9.9%)	YES 3 (12.0%)	0.763
Medication	YES 10 (14.3%)	YES 4 (16.0%)	0.836
Dietary supplements	YES 29 (41.4%)	YES 10 (40.0%)	0.901
Dietary pattern	YES 10 (14.3%)	YES 4 (16.0%)	0.836
Educational level			0.713
Secondary education	18 (25.7%)	7 (28.0%)	
Tertiary education	51 (72.9%)	17 (68.0%)	
Postgraduate education	1 (1.4%)	1 (4.0%)	
MedDiet Score			0.087
Inadequate adherence	11 (15.5%)	8 (32.0%)	
Medium adherence	58 (81.7%)	15 (60.0%)	
High adherence	2 (2.8%)	2 (8.0%)	
Body image inflexibility (BIAAQ-5 ^1^)	57 (80.3%)	7 (28.0%)	<0.001 *
Maternal body image inflexibility (BIAAQ-5 ^1^)	25 (35.2%)	10 (40.0%)	0.778
Paternal body image inflexibility (BIAAQ-5 ^1^)	7 (9.9%)	1 (4.2%)	0.436
Maternal bonding (PBI ^2^)			0.973
Neglectful parenting	13 (18.3%)	4 (16.0%)	
Affectionless control	9 (12.7%)	3 (12.0%)	
Optimal	39 (54.9%)	15 (60.0%)	
Affectionate constraint	9 (12.7%)	3 (12.0%)	
Paternal bonding (PBI ^2^)			0.286
Neglectful parenting	16 (22.5%)	5 (20.0%)	
Affectionless control	16 (22.5%)	2 (8.0%)	
Optimal	29 (40.8%)	16 (64.0%)	
Affectionate constraint	3 (4.2%)	1 (4.0%)	

^1^ BI-AAQ-5 = Body Image Acceptance & Action Questionnaire-5; ^2^ PBI = Parental Bonding Instrument; ^3^ F = female; ^4^ M = male; ^5^ O = Other gender; BMI = body mass index; MedDiet = Mediterranean diet * *p* ≤ 0.05 were considered statistically significant.

## Data Availability

Data are available upon request to the corresponding author.

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
