# Peer review of "The Prevalence of Orthorexia Nervosa among Greek Professional Dancers"

_nutrients, 2023, doi:10.3390/nu15020379_

Round 1

Reviewer 1 Report

The paper "The prevalence of orthorexia nervosa among professional dancers" is very well conducted and contribute to better knowledge about disordered eating attitudes and body shape concerns are prevalent among professional dancers, and appear to be associated with their parental relationship during childhood. Identification of potential ON and development of preventive mechanisms could help to eliminate such concerns and to improve the nutrition of professional dancers.

In my opinion, "greeke dancers" should be added in title, to clarify the reader that this really is in that country. Could be all over the world, but this study focus is in greeke people.

1. What is the main question addressed by the research? The question addressed by the research is to determine the prevalence of orthorexia nervosa (ON) among professional dancers in Greece, and its relationship with nutrition, the body mass index (BMI), body image flexibility and parental bonding.   2. Do you consider the topic original or relevant in the field? Does it address a specific gap in the field? Very important because ortorexia is a new food behavioral problem sometimes under the mask of healthy eating.   3. What does it add to the subject area compared with other published material? The study population was classified into two groups, based on BMI: normal weight and underweight. applied, ON was shown to be significantly correlated with BMI (p=0.006) and body image inflexibility (p<0.001). Parental body image inflexibility was significantly correlated with a low bonding relationship as perceived in childhood. In conclusion, disordered eating attitudes and body shape concerns are prevalent among professional dancers, and appear to be associated with their parental relationship during childhood. Identification of potential ON and development of preventive mechanisms could help to eliminate such concerns and to improve the nutrition of professional dancers.   4. What specific improvements should the authors consider regarding the methodology? What should further controls be considered?   The methodology is clearly addressed; the convinience of sample could be identifided as a limitation of this study.   5. Are the conclusions consistent with the evidence and arguments presented and do they address the main question posed? Yes, the conclusions are in accordance with the results and evidence presented.   6. Are the references appropriate? The references are appropriate, relevant, and in proper number.   7. Please include any additional comments on the tables and figures. A graphical abstract could be added.

Author Response

Reviewer 1

  1. The paper "The prevalence of orthorexia nervosa among professional dancers" is very well conducted and contribute to better knowledge about disordered eating attitudes and body shape concerns are prevalent among professional dancers, and appear to be associated with their parental relationship during childhood. Identification of potential ON and development of preventive mechanisms could help to eliminate such concerns and to improve the nutrition of professional dancers.

Thank you very much for your overall comment

  1. In my opinion, "greeke dancers" should be added in title, to clarify the reader that this really is in that country. Could be all over the world, but this study focus is in greeke people.

The nationality of the dancers is added

  1. What is the main question addressed by the research?The question addressed by the research is to determine the prevalence of orthorexia nervosa (ON) among professional dancers in Greece, and its relationship with nutrition, the body mass index (BMI), body image flexibility and parental bonding.   2. Do you consider the topic original or relevant in the field? Does it address a specific gap in the field? Very important because ortorexia is a new food behavioral problem sometimes under the mask of healthy eating.   3. What does it add to the subject area compared with other published material? The study population was classified into two groups, based on BMI: normal weight and underweight. applied, ON was shown to be significantly correlated with BMI (p=0.006) and body image inflexibility (p<0.001). Parental body image inflexibility was significantly correlated with a low bonding relationship as perceived in childhood. In conclusion, disordered eating attitudes and body shape concerns are prevalent among professional dancers, and appear to be associated with their parental relationship during childhood. Identification of potential ON and development of preventive mechanisms could help to eliminate such concerns and to improve the nutrition of professional dancers.   4. What specific improvements should the authors consider regarding the methodology? What should further controls be considered?   The methodology is clearly addressed; the convinience of sample could be identifided as a limitation of this study.   5. Are the conclusions consistent with the evidence and arguments presented and do they address the main question posed? Yes, the conclusions are in accordance with the results and evidence presented.   6. Are the references appropriate? The references are appropriate, relevant, and in proper number.   7. Please include any additional comments on the tables and figures. A graphical abstract could be added.

We appreciate your comments. A graphical abstract has been prepared and added to the system.

Reviewer 2 Report

Dear authors, 

thank You for presenting a particular article. 

It is well written, with high scientific value, appropriate statistical procedures, and interesting findings.  

The paper has no flaws in scientific work. 

My proposal for enhancement relates to the discussion and conclusion section where some comparison with other authors will be valuable.

In this sense, You can explain how Your work has added scientific value for overall science and enlarged the understanding of the parental influence on future behaviors. 

Author Response

Reviewer 2

Dear authors, 

thank You for presenting a particular article. 

It is well written, with high scientific value, appropriate statistical procedures, and interesting findings.  

The paper has no flaws in scientific work. 

My proposal for enhancement relates to the discussion and conclusion section where some comparison with other authors will be valuable.

In this sense, You can explain how Your work has added scientific value for overall science and enlarged the understanding of the parental influence on future behaviors. 

Thank you for your comment. We appreciate your positive evaluation. Although we have tried to enhance our discussion per your suggestion (line 344), literature in the topic is quite limited, therefore not many additional references could be added.